# Meta-Analysis Suggests That Intron Retention Can Affect Quantification of Transposable Elements from RNA-Seq Data

**DOI:** 10.3390/biology11060826

**Published:** 2022-05-27

**Authors:** Nicolò Gualandi, Cristian Iperi, Mauro Esposito, Federico Ansaloni, Stefano Gustincich, Remo Sanges

**Affiliations:** 1Computational Genomics Laboratory, Area of Neuroscience, Scuola Internazionale Superiore di Studi Avanzati (SISSA), 34136 Trieste, Italy; nicolo.gualandi@gmail.com (N.G.); cristian.iperi@univ-brest.fr (C.I.); mesposit@sissa.it (M.E.); federico.ansaloni@iit.it (F.A.); stefano.gustincich@iit.it (S.G.); 2Central RNA Laboratory, Istituto Italiano di Tecnologia, 16132 Genova, Italy

**Keywords:** transposable elements expression quantification, intron retention, technical bias, RNA-seq, transcriptomic, bioinformatics

## Abstract

**Simple Summary:**

Transposable elements (TEs) are repetitive sequences comprising more than one third of the human genome with the original ability to change their location within the genome. Owing to their repetitive nature, the quantification of TEs results often challenging. RNA-seq is a useful tool for genome-wide TEs quantification, nevertheless it also presents technical issues, including low reads mappability and erroneous quantification derived from the transcription of TEs fragments embedded in canonical transcripts. Fragments derived from TEs are found within the introns of most genes, which led to the hypothesis that intron retention (IR) can affect the unbiased quantification of TEs expression. Performing meta-analysis of public RNA-seq datasets, here we observe that IR can indeed impact the quantification of TEs by increasing the number of reads mapped on intronic TE copies. Our work highlights a correlation between IR and TEs expression measurement by RNA-seq that should be taken into account to achieve reliable TEs quantification, especially in samples characterized by extensive IR, because differential IR might be confused with differential TEs expression.

**Abstract:**

Transposable elements (TEs), also known as “jumping genes”, are repetitive sequences with the capability of changing their location within the genome. They are key players in many different biological processes in health and disease. Therefore, a reliable quantification of their expression as transcriptional units is crucial to distinguish between their independent expression and the transcription of their sequences as part of canonical transcripts. TEs quantification faces difficulties of different types, the most important one being low reads mappability due to their repetitive nature preventing an unambiguous mapping of reads originating from their sequences. A large fraction of TEs fragments localizes within introns, which led to the hypothesis that intron retention (IR) can be an additional source of bias, potentially affecting accurate TEs quantification. IR occurs when introns, normally removed from the mature transcript by the splicing machinery, are maintained in mature transcripts. IR is a widespread mechanism affecting many different genes with cell type-specific patterns. We hypothesized that, in an RNA-seq experiment, reads derived from retained introns can introduce a bias in the detection of overlapping, independent TEs RNA expression. In this study we performed meta-analysis using public RNA-seq data from lymphoblastoid cell lines and show that IR can impact TEs quantification using established tools with default parameters. Reads mapped on intronic TEs were indeed associated to the expression of TEs and influence their correct quantification as independent transcriptional units. We confirmed these results using additional independent datasets, demonstrating that this bias does not appear in samples where IR is not present and that differential TEs expression does not impact on IR quantification. We concluded that IR causes the over-quantification of intronic TEs and differential IR might be confused with differential TEs expression. Our results should be taken into account for a correct quantification of TEs expression from RNA-seq data, especially in samples in which IR is abundant.

## 1. Introduction

Transposable elements (TEs) are repetitive sequences that comprise nearly half of the human genome [1,2]. Their main characteristic is the ability to change their location within the genome and they are found in virtually all eukaryotes [3,4]. TEs are key players in many different biological processes in health and disease [5,6,7]. Their ability to replicate and insert themselves into the genome has been recognized as causing both useful and deleterious effects, and, therefore, cells have evolved different ways to deal with TEs activity [8]. An uncontrolled TEs activation has been related to the onset and progression of different neurological and autoimmune disorders [5]. They can be involved in the generation of new insertions, increasing the number of genomic variations and leading to the onset of disease [9]. They can also impact gene expression by supplying transcriptional regulatory elements such as promoters [10]. An aberrant TEs expression has been found in several neurological and immune diseases such as Alzheimer, Aicardi–Goutières Syndrome, Autism Spectrum Disorders and Multiple Sclerosis as well as in different cancers [11,12,13,14,15]. This evidence suggests the importance of a reliable TEs expression quantification. Consequently, many tools have been so far developed to address this task using data from RNA-seq experiments [16]. In this scenario a reliable quantification of TEs RNA should distinguish their expression as independent transcriptional units from transcription of their sequences as part of canonical transcripts [16]. TEs fragments are indeed embedded within most genes and in particular within intronic sequences. The magnitude of this genomic organization relation is so large that about 90% of all human RefSeq genes contain TEs in their introns [17,18].

Intron retention (IR) is one of the five different alternative splicing (AS) mechanisms. IR occurs when introns, normally removed by the splicing machinery, are maintained in the mature RNA [19]. IR is one of the most abundant AS events in unicellular eukaryotes and plants, in which it accounts for about 60% of all splicing events [20,21]. Despite originally overlooked in animals, IR has recently gained great interest due to its recognized role in gene expression regulation and its association with complex diseases [22,23]. IR is thus currently recognized as a widespread and evolutionary conserved mechanism present in both health and disease [19], affecting many different genes with cell type-specific patterns [24]. IR is a tightly regulated process controlled by many different factors, including splice site strength, intron length, GC content, splicing factor expression, and changes in chromatin conformation [24,25]. This accurate regulation has paved the way to the discovery that IR plays fundamental roles in controlling gene expression by inducing non-sense mediated decay leading to the downregulation of intron-retaining transcripts [26] or by protecting RNAs from degradation by preventing their cytoplasmic export [27]. Recently, it has been reported that IR affects about the 17.4% of expressed genes in human granulocytes, demonstrating its widespread contribution to cell biology [28]. In 2004, Braunschweig and colleagues, using high-coverage polyA RNA-seq data, observed that IR is surprisingly frequent in mammals, affecting transcripts from as many as three-quarters of multi-exonic genes [24]. IR is particularly frequent in blood cells, in which it regulates differentiation and proliferative activity during development and maturation [29,30,31,32,33]. The evidence collected so far confirms that IR is a widespread phenomenon that has relevant outcomes on the transcriptome dynamics in many different cell types.

Interestingly, more than 60% of TEs-derived sequences reside in introns [34]. This relation poses challenges in the quantification of intronic TEs expression as reads coming from retained introns (RI) can bias the detection of overlapping, independent TEs expression [16]. The final outcome might be an incorrect TEs quantification potentially leading to incorrect interpretation of results. This possibility suggests the need of an accurate analysis on the quantitative and qualitative relation between IR and TEs expression quantification.

## 2. Materials and Methods

### 2.1. Data Collection and Pre-Processing

To study the relation between IR and TEs quantification we took advantage of the large cohort of the Geuvadis dataset [35]. Geuvadis project sequenced polyA mRNA from lymphoblastoid cell lines derived from 464 healthy individuals from the 1000 Genomes Project, generating a dataset of uniformly processed RNA-seq data from multiple human populations with high-quality sequences. The NA18861 sample has been removed from the analysis because it represents an extreme outlier concerning the number of retained introns. In addition to the Geuvadis dataset, we used two additional datasets to test and report practical examples of the influences of IR on TEs quantification. The former is composed by polyA RNA-seq data from human neural progenitor cells (hNPCs) harboring a homozygous deletion of the *DNMT1* gene, leading to a global loss of DNA CpG-methylation and causing a significant increase in autonomous TEs expression [36]. This dataset represents a model to test whether the validated dysregulation of TEs can be correctly identified in RNA-seq data where differential IR is not abundant. The latter is composed by RNA-seq data from SDE2 knock-down HeLa cells [37]. Data are declared as polyA RNA; however, the kit used to generate the sequencing library suggests that the RNA has to be considered as total RNA with ribosomal depletion. SDE2 is a human RNA binding protein and a trans-acting, splicing-associated factor required for efficient RNA processing. The silencing of SDE2 in human cell lines has been associated with widespread changes in alternative splicing, with increased IR being the most frequent event [37]. This dataset represents an ideal model to explore TEs measurements in presence of differential IR. To simulate the scenario of differential IR between two analyzed groups we also performed a comparison of four samples with a high number of retained introns (H-IR) and four samples with a lower number of retained introns (L-IR) extracted from the Geuvadis project. Groups were defined by retrieving the four samples with the highest number of RIs and the four samples with the lowest number of RIs, after excluding samples from the 2% tails of the distribution to avoid extreme outliers. The Appendix A summarizes the information about the different datasets used in this study. Raw fastq files were downloaded from the ENA-EBI database (accession codes: PRJEB3366, PRJNA420729, and PRJNA599420) and the quality of the reads was assessed using FastQC [38,39].

### 2.2. IR Quantification and Differential IR Analysis

Reads were mapped to the reference genome (version hg19, Ensembl) using STAR (v 2.7.0) with default parameters [40,41]. For the Geuvadis dataset, different fastq files from the same samples were merged at the mapping step to increase the sensitivity of IR detection as the number of detectable RIs is influenced by the number of mapped reads. IR was quantified using IRFinder [42] with default parameters. The IRFinder main metrics are represented by the IRratio that quantifies the portion of transcriptional activity traversing a given intron not removed by the splicing mechanisms. It is therefore a measure representing, for a given gene, the percentage of transcripts retaining the introns with respect to the total number of transcripts. The number of RIs per sample was defined as the number of introns with an IRratio >= 0.1 and without any overlap with annotated features (the “clean” annotation in the IRFinder output), such as exons, which can confound the correct IRratio quantification. The set of commonly used RIs for the randomization analysis of the Geuvadis dataset is composed of 2789 introns, defined as introns retained in at least the 65% of samples (Appendix A). This approach is justified by the fact that IR is a highly conserved and regulated mechanism with a cell type-specific pattern [24]. The genomic coordinates of the quantified introns were extracted from the BED file generated by IRFinder. Differential IR analysis was carried out using DESeq2, according to the IRFinder manual for datasets with more than 3 biological replicates. Significant results were considered as having an FDR adjusted *p*-value < 0.05 and a |log2(fold change)| > log2(1.5).

### 2.3. Transposable Elements Expression Quantification and Differential Expression Analysis

Locus-specific TEs expression was quantified using SQuIRE [43] with default parameters and raw counts were normalized using the DESeq2 default median of ratios approach [44] (using the counts function with the normalized = TRUE parameter) to account for both TEs dispersion and different sequencing depth across samples. The set of transcribed TEs was defined in the Geuvadis dataset as TEs showing an expression level of at least 5 normalized counts in at least the 65% of samples, comprising 36,383 TEs, and was further divided into intronic and intergenic using BEDtools utilities (Appendix A) [45]. Intronic TEs were defined as TEs overlapping at least one intron, while intergenic TEs were defined as TEs without overlap with any annotated gene. The gene annotations were retrieved from Ensembl Biomart by selecting all coding and non-coding genes [46]. Differential TEs expression analysis was performed using DESeq2. Significant results were considered as having an FDR adjusted *p*-value < 0.05 and a |log2(fold change)| > log2(2).

### 2.4. Statistical and Genomic Analysis

Statistical analysis was conducted in R and Python. In particular, Pearson’s correlation (*scipy* Python module) between normalized TEs expression and the hosting intron IRratio was performed for every intronic TE (Appendix A). TEs and intron pairs were defined using BEDtools intersect. *p*-values were then corrected for multiple tests using the FDR method. Significant correlations were defined as correlation with an FDR adjusted *p*-value < 0.05. To test the relation between TEs quantification and IR that showed even low but reproducible evidence of retention, only introns with an IRratio > 0 in at least 50% of samples were used in this test. The genomic location of upregulated TEs was assessed using the annotatr R package and the hg19_basicgenes and hg19_genes_intergenic regions. These annotations contain regions for 1to5kb, promoters, 5UTRs, exonintronboundaries, intronexonboundaries, exons, introns, 3UTRs, and intergenic. In particular, the 1–5 Kb is the region located from 1 to 5 Kb upstream of the transcription start site (TSS); the promoter region is defined as the region from 0 to 1 Kb upstream the TSS; the intergenic region does not contain all the previous annotations. If an element overlaps more than one annotation, the results were prioritized maintaining only the first overlap with respect to the reported order. The enrichment in different genomic regions was tested by comparing the real number of elements overlapping a certain region with the mean of 1000 randomizations of the same number of elements taken from an unbiased background of transcribed TEs and by computing the Z-score. Randomizations were performed using the resampleRegions function from the regioneR package [47]. Common reads between TEs in RIs and intergenic TEs were counted based on read names. Briefly, TEs located in RIs were identified by intersecting the BED file containing the TEs genomic location with the BED file of RIs. Read names of reads mapped in those regions were then extracted using samtools. The same procedures were repeated to identify the read names of reads mapped in intergenic TEs by retrieving the genomic location of intergenic TEs as TEs that do not overlap any gene annotation (BEDtools intersect -v), then read names were extracted using Samtools. The common reads were identified using the comm Bash command that can be used to identify the common names between the 2 lists of read names (files were previously sorted by name). The correlation between the number of RIs and the number of transcribed intronic TEs was performed using the cor.test function in R. The number of RIs was assessed as previously described and normalized on the total number of mapped reads to allow inter-sample comparisons while the number of transcribed TEs per sample was defined as the TEs with counts >= 5 after DESeq2 normalization.

## 3. Results

### 3.1. Introns Contain a Large Fraction of TEs That Can Be Measured as Transcribed in RNA-Seq Experiments

To explore the phenomenon of IR at a whole transcriptome level on a comprehensive dataset for the human species we took advantage of the Geuvadis project data, composed of RNA-seq from 463 samples. We identified RIs by using IRFinder [35,42]. The tool was able to measure the IRratio of 188,138 introns that do not overlap any exon annotation, with a mean of 6322 RIs per sample (3.36% of all measurable introns, Figure 1A), which is in agreement with numbers reported in the literature [28]. Progresses in human genome annotation and TEs sequences identification have revealed that TEs fragments are embedded within most genes and, in particular, inside intronic sequences [48]. The magnitude of this general genomic organization is so important that about 90% of all human RefSeq genes contain TEs in their introns [17,18]. This led us to hypothesize that IR can have an impact on intronic TEs quantification by increasing the number of reads mapped on intronic TEs. In fact, the high degree of genomic overlap among introns and embedded TEs poses challenges in correctly measuring TEs expression using RNA-seq data because reads, coming from RIs, might represent a confounding factor (Figure 1C) [16]. In order to determine the genomic localization of TEs that are reproducibly identified in the analyzed RNAseq reads, we selected the TEs showing expression in at least the 65% of analyzed samples and considered them as transcribed TEs (36,383 elements). These might be both autonomously transcribed TEs as well as TEs fragments transcribed as part of other transcriptional units. We compared the genomic distribution of the selected transcribed TEs to 1000 randomizations of their locations and observed a significant enrichment of transcribed TEs in gene bodies, beginning at the core promoter (1 kb upstream of the TSS). This also results in intronic regions being slightly, but significantly, enriched, with respect to randomizations and introns themselves containing the highest fraction of transcribed TEs. Conversely, intergenic regions are strongly and significantly depleted of the identified transcribed TEs (Figure 1B). The significant enrichment of transcribed TEs inside all regions of the gene body (UTRs, introns, and exons) is expected because exonized TE fragments represent an important fraction of the human transcriptome [49,50]. The transcription of genes containing these TE sequences might therefore produce reads mapping on these exonized TE fragments. However, introns should be removed from mature transcripts and reads mapping on intronic TEs should not be expected in mature transcripts unless the mapped TEs are autonomously transcribed or in cases of intron retention. The presence of intronic reads in polyA RNA-seq data has been proposed to be a random bias produced during library preparation. However, they should not be reproducible among different samples, which rules out the presence of this bias in our selection of reproducibly transcribed TEs.

### 3.2. Intron Retention Can Introduce a Bias in Intronic TEs Quantification

To identify a possible relationship between intronic TEs transcription and IR, the per-sample mean expression of TEs located in RIs was compared with the mean expression of the same number of TEs located in randomly selected introns (1000 randomization). The result reported in Figure 2A shows that TEs located in RIs result in being expressed at a significantly higher level with respect to TEs located in randomly selected introns, suggesting that IR can result in the increased quantification of intronic TEs from RNA-seq data. Then, to investigate the existence of a quantitative relation between IR and intronic TEs quantification at a genome-wide level, correlation analysis between the expression levels of intronic TEs and the IRratio of the hosting introns was performed. Our results show that there are 1,209,243 measurable intronic TEs fragments located in 65,147 measurable introns (Appendix A). Correlations analysis show that the level of expression of the majority (74.1%) of intronic TEs significantly and positively correlates with the IRratio of the hosting intron (Figure 2B,C). On the other hand, there is a low percentage (0.4%) of intronic TEs showing a negative and significant correlation with IRratio of the host intron (Figure 2B). The high percentage of significantly positive correlations between TEs measurements and the hosting intron IRratio supports the idea that IR has an impact on intronic TEs quantification and that an increased IR is associated with an increased measure of the expression of hosted TEs. Moreover, we have also observed that the set of commonly RIs is significantly enriched for containing transcribed TEs with respect to randomly selected introns (Figure 2D) (Z-score = 65). The presence of TEs sequences inside introns might impact the efficiency of intron splicing influencing the IR level [22]. Considering this possibility, we have investigated whether the set of common RIs is enriched to contain TEs sequences with respect to randomly chosen introns. By answering this question, we can rule out a possible association between TEs sequences and IR in the analyzed dataset The number of common RIs overlapping at least one TE, regardless of its expression, were counted and compared with the mean of 1000 randomizations of randomly chosen introns. As shown in Figure 2E, the set of common RIs is depleted (Z-score = −12) of TE sequences with respect to randomly chosen introns. This result indicates that the presence of TEs inside introns is not associated with an increase in the level of IR in the analyzed samples.

In order to confirm that a global increase in IR might induce a consequent increase in the number of transcribed intronic TEs, the number of RIs was correlated with the number of transcribed intronic TEs in each sample analyzed. The Figure 3A shows a significant correlation (R = 0.71 and *p*-value = 1.24 × 10^−73^) between the number of RIs and the number of transcribed intronic TEs in each sample. This result again suggests that an increase in IR might cause an increased number of transcribed intronic TEs.

TEs are characterized by a high sequence similarity, at least among members of the same family, so we should consider that an increase in IR can also influence the measurements of other non-intronic TEs [51]. In this scenario, reads coming from RIs can map on both intronic and intergenic TEs, therefore possibly impacting the quantification of intergenic TEs [16]. To further investigate this matter, the number of multi-mapping reads mapped on TEs located in both RI and intergenic TEs in each sample was measured. These were not significantly correlated with the number of transcribed intergenic TEs in each sample (R = 0.03 *p*-value = 0.5) (Figure 3B). The absence of a significant correlation suggests that IR has had not a major impact on intergenic TEs quantification. We also observed that only a small fraction of reads is shared by both TEs in RI and intergenic TEs (Figure 3C). These results suggest that IR does not have a strong impact on the quantification of intergenic TEs and therefore the quantification of intergenic TEs might be considered more reliable even in samples with a high level of IR.

### 3.3. IR and TEs Quantification: Validations Using Alternative Datasets

#### 3.3.1. TEs Quantification Is Biased by IR in Dataset Characterized by Differential IR between Two Groups

In order to simulate the scenario of differential IR between two analyzed groups, we decided to compare four samples with a high number of RIs (H-IR) and four samples with a low number of RIs (L-IR) extracted from the Geuvadis project. As expected, the differentially RIs analysis revealed 323 introns with increased IR in the H-IR group and no introns with increased IR in the L-IR group (Figure 4A). This experimental setup provided a good model to explore the analysis of differential TE expression. A global TEs deregulation was observed, with 1175 upregulated TEs and 1642 downregulated TEs in the H-IR group with respect to the L-IR (Figure 4B). To further investigate the differential TEs expression analysis in a context characterized by increased IR between the 2 groups, we studied the distribution of the upregulated TEs with respect to the gene structure. As shown in Figure 4C, the majority of upregulated TEs are located in introns. Moreover, randomizations analysis indicates that the number of upregulated intronic TEs is higher than expected by chance suggesting that the increased IR have introduced a bias in the correct measurement of intronic TEs expression (Z-score = 27.5) (Figure 4C). Interestingly, we also observed a significant depletion of upregulated TEs in intergenic regions confirming that IR has not a major impact on intergenic TEs quantification (Z-score = −11.4) (Figure 4C).

#### 3.3.2. Global TEs and IR Quantification Are Not Biased in a Dataset Characterized by True Autonomous TEs Expression

In order to understand whether TEs quantification is biased by IR in any type of biological context and to rule out the possibility that IR can be biased in case of TEs upregulation, the genomic distribution of upregulated TEs was investigated in an RNA-seq dataset characterized by strong and validated TEs upregulation [36]. The chosen dataset is composed by RNA-seq data from human neural progenitor cells (hNPCs) harboring a homozygous deletion of the *DNMT1* gene. *DNMT1* encodes for the DNA methyltransferase that play a key role in DNA methylation maintenance. In this cellular model the authors have observed a global loss of DNA CpG-methylation leading to autonomous TEs upregulation [36]. This dataset represents a good model to understand the reliability of TEs measurements in presence of experimentally validated, autonomous upregulation of TEs and its impact on IR measurements. The differential TEs expression analysis identified 2804 upregulated and 118 downregulated TEs in *DNMT1 ^-/-^* cells with respect to WT controls (Figure 4E). In particular, the majority of them are intergenic while ~750 upregulated TEs are intronic (Figure 4F). To explore the genomic distribution of the upregulated TEs and to quantify their relations with introns, a randomization analysis was performed. In contrast to what previously observed in the Geuvadis dataset (Figure 4C), these results show that upregulated TEs are significantly enriched within intergenic regions with respect to the mean of 1000 randomizations (Z-score = 58.5) (Figure 4F). Accordingly, we also observe a concomitant depletion of upregulated TEs in intronic regions (Z-score = −18.1) (Figure 4F). This result highlights that, in this dataset, upregulated TEs are composed by a significantly higher proportion of intergenic TEs while the number of upregulated intronic TEs is significantly lower than expected by chance. Moreover, analysis aimed at detecting differentially retained introns between DNMT1^-/-^ and control cells revealed that there are no introns characterized by differential retention between the two groups of samples (Figure 4D). The absence of differentially retained introns between the two groups and the depletion of upregulated TEs in intronic regions, confirm the absence of a systematic bias introduced by TEs expression on IR. We conclude that in datasets characterized by autonomous TEs upregulation and absence of differential IR, the TEs quantification is less subjected to biases and that TEs expression has no impact on IR quantification.

#### 3.3.3. TEs Quantification Is Biased by IR in a Datasets Characterized by Strong Differential IR

To provide an example of how differential IR can bias a correct differential expression analysis of TEs, the genomic distribution of upregulated TEs was investigated in an alternative dataset characterized by a strong differential IR between two groups of samples [37]. The dataset is composed by RNA-seq data from HeLa cells treated with siRNA targeting the SDE2 gene. SDE2 encodes for an RNA-binding protein involved, among others, in AS process. SDE2 silencing has previously resulted in global changes in AS with an increase in IR [37]. Differential IR analysis of this dataset revealed 9148 introns with increased IR in SDE2 knock-down samples (Figure 4G). This confirms that the dataset is characterized by an extensive deregulation of IR. Differential TEs expression analysis resulted in more than 7000 upregulated TEs in SDE2 knock-down cells with respect to controls compatible with a global upregulation of TEs expression (Figure 4H). However, the vast majority of them are located in introns with only few upregulated intergenic TEs (Figure 4I). Upregulated TEs result significantly enriched within intronic regions with respect to the mean of 1000 randomizations (Z-score = 67.1) (Figure 4I). Accordingly, our analysis also shows a concomitant depletion of upregulated TEs in intergenic regions (Z-score = −29.5) (Figure 4I). This result highlights that, in this experiment, upregulated TEs are composed by a significantly higher proportion of intronic TEs while the number of upregulated intergenic TEs is significantly lower than expected. The presence of differentially RIs combined with the enrichment of upregulated TEs in intronic regions confirms that an increased IR can induce an over-estimation of the number of upregulated intronic TEs.

## 4. Discussion

Because of the increasingly recognized role of TEs in health and disease [5,6,7], accurate measurements of TEs expression represent a crucial part of transcriptomics studies [16]. Being in its infancy and because of the intrinsic repetitive nature of these elements, this type of analysis faces several difficulties, one of the most important is the low reads mappability of RNAseq data that prevents the unambiguous assignment of a large fraction of reads derived from transcribed TEs [16]. This problem is worsened by random mutations and other forms of TE sequence alterations that can prevent a correct mapping of TE-derived reads onto the reference genome [16]. Our study demonstrates that, among the aforementioned confounding factors, IR can contribute to TEs over-quantification. First, we showed that IR is a widespread mechanism affecting about 6000 introns in a given lymphoblastoid cell line. In this scenario, we demonstrated evidence suggesting that IR is able to affect the correct and independent TEs quantification by increasing the number of reads assigned to intronic TEs. As a consequence, we reported that TEs located in RI are characterized by a systematic higher expression level with respect to TEs located in randomly selected introns. Using correlation analysis, we showed that more than 74% of intronic TEs quantification positively correlates with the level of IR of the hosting intron, while only the 0.4% has a negative correlation, thus adding further support to the idea that IR affects a correct measurement of intronic TEs. Moreover, the number of transcribed intronic TEs correlates with the number of RIs. Following randomization analysis, using a set of common RIs and transcribed TEs shared by at least the 65% of samples, it was proven that RIs are enriched by transcribed TEs, while a matched-size set of non-RIs does not show such enrichment. These results again suggest that IR affects TEs measurements, whereas the presence of intronic TEs sequences does not impact IR, at least in the analyzed dataset. The repetitive nature of TEs have led us to hypothesize that IR can also impact the quantification of intergenic TEs that share a high sequence similarity with TEs located in RIs. However, our results showed that, on average, only the 5% of reads mapped on TEs that are located in RIs are also mapped on intergenic TEs. These results suggest that the expression levels of intergenic TEs might be considered more reliable and not generally affected by IR, although further analyses are needed to assess its potential impact at a locus-specific level. Finally, we reported on three practical examples used to explore how IR can affect TE measurements and differential expression analysis in publicly available datasets. Between the analyzed groups, we observed that datasets characterized by a differential IR had a significant enrichment of upregulated TEs in intronic regions and a concomitant depletion from intergenic regions in the samples affected by IR. Our explanation is that the increased number of reads derived from RIs systematically increases the quantification of intronic TEs, resulting in their over-quantification. We also reported that in datasets characterized by true autonomous TEs upregulation and the absence of strong IR, the differential TEs expression analysis does not result in bias and it does not affect correct IR quantification.

Our study suggests that results from the differential expression analysis of TEs should be taken with care in biological contexts where differential IR happens. Nowadays, the perfect tool for TEs quantification probably does not exist because of the repetitive nature of TEs and because portions of their sequences have frequently been exapted during evolution [3,10,52]. When analyzing TEs, it is hard to develop methods characterized by high sensitivity while maintaining high specificity. Therefore, it has been suggested that an ensemble approach, combining different methods, may be the solution to increase sensitivity when analyzing TEs using short reads [53]. Many excellent and sophisticated approaches have nevertheless been developed to assess a relatively reliable quantification of TEs expression [16]. Many of them are focused on limiting low mappability issues and to find a way to reliably assign multi-mapping reads. SQuiRE uses an expectation-maximization algorithm to redistribute multi-mapping read fractions to quantify TE expression [43]. ERVmap and TeXP represent two alternative tools, based on the construction of curated database of autonomously transcribed elements or the building of mappability signatures to identify autonomous LINE-1 activity, respectively [54,55]. Other approaches have instead tried to remove noise from gene transcription by removing reads mapped on both TEs and genes [56]. These approaches have greatly improved TEs quantifications but, to our knowledge, none of these have been specifically designed to tackle the bias introduced by IR. A simple strategy would be to remove reads mapped on intronic regions during the quantification. This would eliminate the potential bias introduced by IR but would lead to the discarding of many reads and in turn lower the power of the approach and the significance of the analysis. Another solution would be to remove only reads derived from RIs before the quantification step of TEs. With this strategy, one would remove the over-quantification coming only from RIs and avoid discarding other TEs-specific reads mapped in non-RIs. An interesting solution proposed by Chung et al. consists in reducing the read count of intronic TEs proportionally to the intronic flanking regions [57]. The spread and acquisition of third generation sequencing strategies characterized by the production of long reads will probably constitute the greatest advantage that will solve, or at least strongly mitigate, the problems highlighted herein. While second-generation sequencing technologies have brought advantages in transcriptome analysis, one of their main limits is the short read length, which, for non-unique sequences, can prevent an unambiguous mapping determining a consequent confounding effect on downstream analyses, such as the identification of AS isoforms and TEs autonomous expression. Third-generation sequencing technologies represent an exciting perspective for the identification and validation of both AS events and TEs quantification. Among third-generation technologies, PacBio and Oxford Nanopore have gained popularity because of their capability to produce very long reads. Increased read length represents one of the most promising features of these technologies. Because of the much longer read lengths these can indeed help to reconstruct the precise structure of a single transcript isoform using a single read without the need to assemble short reads. This can facilitate the identification and quantification of splicing isoforms and help to distinguish between passive and autonomous TEs transcription. In 2021, Lee and colleagues used Oxford Nanopore sequencing to characterize AS in Apicomplexan Parasites. They identified widespread AS with IR being one of the most abundant events [58]. This confirms that long reads are indeed useful for the identification of IR. Even if these technologies are of great interest, it is important to keep into account that they are characterized by lower throughput, higher error rate, and higher cost per base, with respect to short reads sequencing technologies and therefore their use is still limited at the moment.

In general, we advise the careful evaluation of the existence of differential IR every time the differential expression analysis of TEs is of interest. This is particularly important in comparing diseased and healthy samples because differential IR has started to be reported in many diseases [59,60,61,62] and could lead to the wrong measurements of differential expression of TEs, misleading the interpretation of results. This work highlights the need for careful and appropriate quantification and differential expression analysis of TEs and suggests giving significant attention to the extent of IR in samples under consideration. Our study is based exclusively on the meta-analysis of public data from published research and adds strong support to the idea that IR can prevent an accurate and independent TE quantification introducing a bias towards intronic TEs. We have selected several studies in which the expression of TEs and/or IR had been carefully validated. However, specific wet-lab experiments performed on specific introns/TEs pairs with alternative techniques, such as quantitative and/or digital PCRs, will ultimately confirm the results herein presented beyond any reasonable doubt.

## 5. Conclusions

TEs have started to be recognized as important players in health and disease; thus, an accurate quantification of their expression has become fundamental in many research projects. Accurate TE quantification faces a variety of confounding factors, among which low reads mappability is one of the most impactful. In this study, we added support to the idea that IR has an influence on accurate TEs quantification. Indeed, reads deriving from RIs might confound the quantification of intronic TEs by increasing the number of reads mapped on TEs, leading to their over-quantification. IR was considered a consequence of aberrant splicing, but it has recently gained attention due to the discovery of biological processes in which this phenomenon plays a fundamental role. Moreover, IR seems to be more frequent than expected, especially in cells from blood or the nervous system. The possibility that IR can bias an accurate and independent quantification of TEs suggests the need to inspect the extent of IR before interpreting any data on TEs expression in samples characterized by high IR.

## Figures and Tables

**Figure 1 biology-11-00826-f001:**
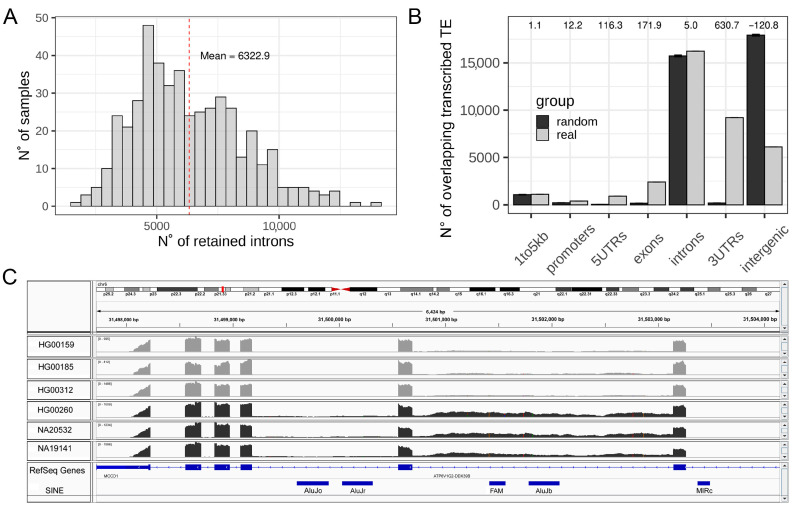
Exploration of IR and TEs in RNA-seq data from Geuvadis. (**A**) Distribution of the number of retained introns per samples in the Geuvadis dataset. (**B**) Genomic distribution of transcribed TEs (light grey) compared to the mean of 1000 randomization (dark grey). Z-scores are reported as numbers on the top of the plot. Error bars represent the standard deviation. (**C**) Integrative Genomics Viewer (IGV) screenshot of an intron characterized by increase retention in three samples colored in dark grey with respect to the light grey samples. This intron contains two annotated repeated elements, which could result in being differentially expressed if analyzed without checking the retention of their hosting intron.

**Figure 2 biology-11-00826-f002:**
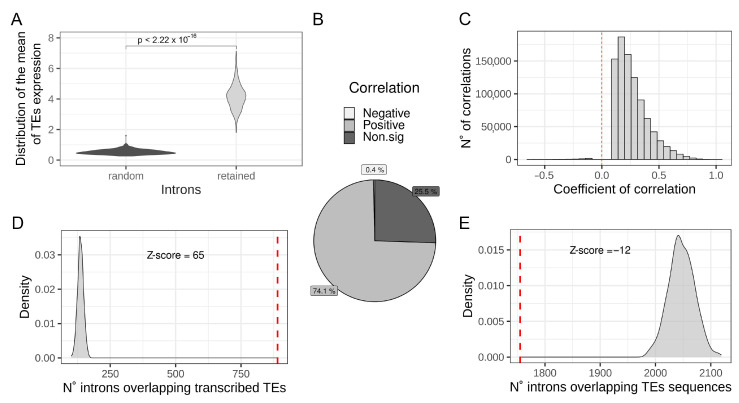
Intron retention might increase intronic TE measurements. (**A**) Distribution of the mean expression of TEs located in retained introns (light grey) compared to the mean expression of TEs located in randomly chosen introns (dark grey). *p*-value is reported on the top of the plot. (**B**) Pie chart reporting the percentage of positive, negative, and non-significant correlations between TE expression and the hosting intron’s IRratio. (**C**) Distribution of significant (adjusted *p*-value < 0.05) coefficients of correlation between TE expression and hosting intron’s IRratio. (**D**) Comparison between the number of commonly retained introns containing at least one transcribed TE (red dashed line) in the Geuvadis dataset with the mean of 1000 randomizations of randomly chosen introns (light grey distribution). Z-score is reported on the top of the plot. **E** Comparison between the number of commonly retained introns containing at least one TE fragment (red dashed line) in the Geuvadis dataset compared to the mean of 1000 randomizations of randomly chosen introns (light grey distribution). Z-score is reported on the top of the plot.

**Figure 3 biology-11-00826-f003:**
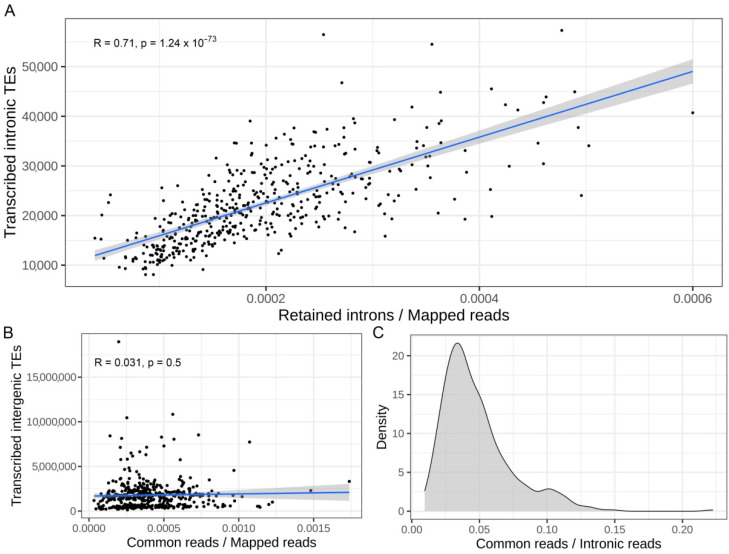
Increased intron retention correlates with an increased number of expressed intronic TEs. (**A**) Correlation between normalized retained introns counts and the number of transcribed intronic TEs per sample. On the top of the plot the coefficients of correlation (R) and the *p*-value are reported. (**B**) Correlation between normalized number of reads in common between TEs in RI and intergenic TEs and the number of transcribed intergenic TEs per samples are reported. On the top of the plot the coefficients of correlation R and the *p*-value are shown. (**C**) Distribution of the number of common multi-mapping reads among TEs located in retained introns and intergenic TEs per sample. The number is reported as a fraction with respect to the total number of reads mapped in TEs located in retained introns.

**Figure 4 biology-11-00826-f004:**
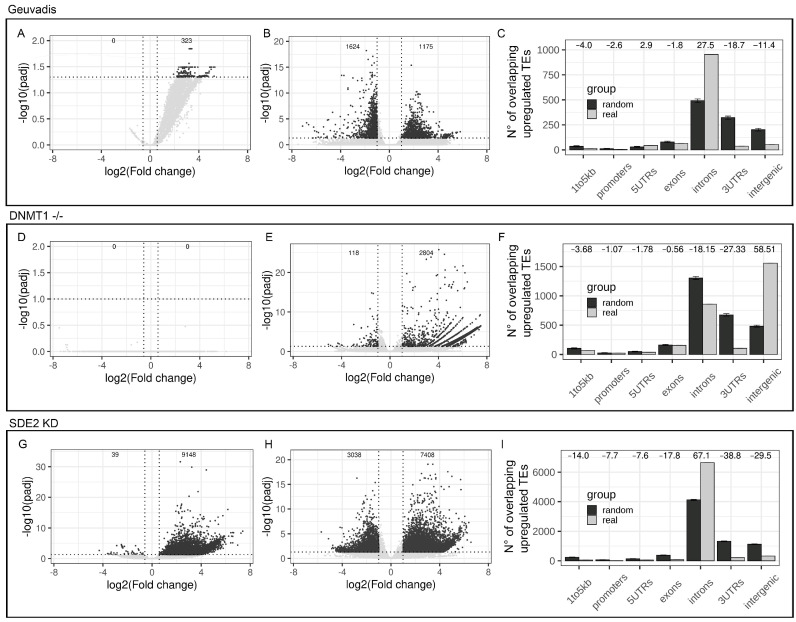
Differential intron retention between groups biases a correct TEs differential expression analysis. (**A**) Volcano plot reporting the number of deregulated introns in IR-High vs. IR-Low samples from Geuvadis dataset. Significant results are reported in dark grey (**B**) Volcano plot reporting the number of deregulated TEs in IR-High vs. IR-Low samples from Geuvadis dataset. Significant results are reported in dark grey. (**C**) Genomic distribution of upregulated TEs (light grey) in IR-High with respect to IR-Low samples compared to the mean of 1000 randomizations (dark grey). Z-score are reported as numbers on the top of the plot. Error bars represent the standard deviation. (**D**) Volcano plot reporting the number of deregulated introns in DNMT^-/-^ samples vs. WT control. Significant results are reported in dark grey. (**E**) Volcano plot reporting the number of deregulated TEs in DNMT^-/-^ samples vs. WT control. Significant results are reported in dark grey. (**F**) Genomic distribution of upregulated TEs (light grey) in DNMT^-/-^ samples, with respect to WT controls, compared to the mean of 1000 randomizations (dark grey). Z-scores are reported as numbers on the top of the plot. Error bars represent the standard deviation. (**G**) Volcano plot reporting the number of deregulated introns in SDE2 knock-down cells with respect to control cells. Significant results are reported in dark grey. (**H**) Volcano plot reporting the number of deregulated TEs in SDE2 knock-down cells with respect to controls samples. Significant results are reported in dark grey. (**I**) Genomic distribution of upregulated TEs (light grey) in SDE2 knock-down cells compared to the mean of 1000 randomizations (dark grey). Z-scores are reported as numbers on the top of the plot. Error bars represent the standard deviation.

## Data Availability

RNA-seq data from Geuvadis project used in this study are available in the ENA-EBI repository (project name: PRJEB3366) https://www.ebi.ac.uk/ena/browser/view/PRJEB3366 (accessed on 5 December 2019). RNA-seq data from DNMT1 knock out project used in this study are available in the ENA-EBI repository (project name: PRJNA420729) https://www.ebi.ac.uk/ena/browser/view/PRJNA420729 (accessed on 19 November 2020). RNA-seq data from SDE2 knock down cells used in this study are available in the ENA-EBI repository (project name: PRJNA599420) https://www.ebi.ac.uk/ena/browser/view/PRJNA599420?show=reads accessed on (22 January 2021).

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
