# Peer review of "Meta-Analysis Suggests That Intron Retention Can Affect Quantification of Transposable Elements from RNA-Seq Data"

_biology, 2022, doi:10.3390/biology11060826_

Round 1

Reviewer 1 Report

Abstract :

The aim of this paper is to demonstrate that intron retention (IR) has an influence on accurate TEs quantification.

Because of the increasingly recognized role of TEs in health and disease, accurate measurements of TEs expression represent a crucial part of transcriptomics studies. Because of the intrinsic repetitive nature of these elements, this type of analysis faces with several difficulties, one of the most important is low reads mappability of RNAseq data that prevents unambiguous assignment of a large fraction of reads deriving from transcribed TEs.

This study shows that reads deriving from RI might confound the quantification of intronic TEs by increasing the number of reads mapped on TEs leading to their over-quantification. In fact, the authors first showed that IR is a widespread mechanism affecting around 6000 introns in a given lymphoblastoid cell line and that IR is able to affect the correct and independent TEs quantification by increasing the number of reads assigned to intronic TEs. They reported that TEs located in retained introns are characterized by a systematic higher expression level with respect to TEs located in randomly selected introns. Using correlation analysis, they shown that more than 78% of intronic TEs quantification positively correlates with the level of IR of the hosting intron, thus confirming that IR affects a correct measurement of intronic TEs. Finally, they reported three practical examples to explore how IR can affect TEs measurements and differential expression analysis in public available datasets. They observed that datasets characterized by a differential IR between the analyzed groups have a significant enrichment of upregulated TEs in intronic regions and a concomitant depletion from intergenic regions in the samples group affected by IR.

So, this study highlights that IR can bias an accurate and independent quantification of TEs suggests the need to inspect the extent of IR before interpretating any data on TEs expression in samples characterized by high IR.

General comments :

In this article, the authors highlight that intron retention (IR) has an influence on accurate TEs quantification. The introduction provides sufficient background, and the methods are adequately described except for a few details described in specific comments. The results are clearly presented with robust statistics, and the discussion is well argued.

Specific comments:

In materials and methods, some information is missing. For example, in “2.1. data collection”, it would be nice to indicate whether it is single or paired reads and the size of the reads.

In “2.2 IR quantification”, the authors used FDR adjusted p-value < 0,1 which seems a little bit high because we often use 0,05, similarly in part “2.4 Statistical”. Can the authors explain their choice?

In 2.4, can the authors explain precisely what “1to5kb” means?

In the results, “3.2 Intron retention can introduce…”, line 270, can the authors specify under which mapping conditions they carried out the analyses? default settings?

In the discussion, it would be nice if the authors discuss new technologies like nanopore, Pacbio.

Small corrections:

Line 14: There is written “map pability” instead of “mappability”.

Line 152: In the sentence, “Between TEs quantification and IR in introns”, inevitably intron retention is in the introns, isn’t it?

Author Response

Please see the attachment. Many Thanks.

Reviewer 2 Report

The Ms entitled ‘Intron retention can affect quantification of transposable elements from RNA-seq data’ is nicely written. The study is interesting and nicely executed. However a few confirmatory experiments should be done.

  1. The differential expression is based on RNA seq data only, which usuaaly creates biased among the isoforms. Authors should revalidate the expression of certain TEs transcript isoforms with and without introns by quantitative RT PCR using primers from unique regions.
  2. It is not clear how many RNA seq data were used for expression analysis? Are those data were generated in replicates? Do they represents diverse stages of development or any treatment? These should be clarified and explained that will increase the relevance of study.

Author Response

Please see the attachment. Many Thanks.

Round 2

Reviewer 2 Report

Ms has been improved. However some minor language edit is required before acceptance.